# Molecular Classification and Overcoming Therapy Resistance for Acute Myeloid Leukemia with Adverse Genetic Factors

**DOI:** 10.3390/ijms23115950

**Published:** 2022-05-25

**Authors:** Daisuke Ikeda, SungGi Chi, Satoshi Uchiyama, Hirotaka Nakamura, Yong-Mei Guo, Nobuhiko Yamauchi, Junichiro Yuda, Yosuke Minami

**Affiliations:** 1Department of Hematology, National Cancer Center Hospital East, Kashiwa, Chiba 277-8577, Japan; ikeda.daisuke@kameda.jp (D.I.); schi@east.ncc.go.jp (S.C.); suchiyam@east.ncc.go.jp (S.U.); hirotnak@east.ncc.go.jp (H.N.); ekaku@east.ncc.go.jp (Y.-M.G.); noyamauc@east.ncc.go.jp (N.Y.); jyuda@east.ncc.go.jp (J.Y.); 2Department of Hematology, Kameda Medical Center, Kamogawa 296-8602, Japan

**Keywords:** ELN classification, AML, *FLT3*-ITD with wild-type *NPM1*, *DEK-NUP214* fusion, *KMT2A* rearrangement, *BCR-ABL1* fusion, haploinsufficiency of *GATA2* and mis-expression of *MECOM*, RUNX1 mutation, *ASXL1* mutation, *TP53* mutation, complex karyotype, menin, anti-CD47 antibody

## Abstract

The European LeukemiaNet (ELN) criteria define the adverse genetic factors of acute myeloid leukemia (AML). AML with adverse genetic factors uniformly shows resistance to standard chemotherapy and is associated with poor prognosis. Here, we focus on the biological background and real-world etiology of these adverse genetic factors and then describe a strategy to overcome the clinical disadvantages in terms of targeting pivotal molecular mechanisms. Different adverse genetic factors often rely on common pathways. *KMT2A* rearrangement, *DEK-NUP214* fusion, and *NPM1* mutation are associated with the upregulation of *HOX* genes. The dominant tyrosine kinase activity of the mutant FLT3 or BCR-ABL1 fusion proteins is transduced by the AKT-mTOR, MAPK-ERK, and STAT5 pathways. Concurrent mutations of *ASXL1* and *RUNX1* are associated with activated AKT. Both *TP53* mutation and mis-expressed *MECOM* are related to impaired apoptosis. Clinical data suggest that adverse genetic factors can be found in at least one in eight AML patients and appear to accumulate in relapsed/refractory cases. *TP53* mutation is associated with particularly poor prognosis. Molecular-targeted therapies focusing on specific genomic abnormalities, such as *FLT3*, *KMT2A*, and *TP53*, have been developed and have demonstrated promising results.

## 1. Introduction

Patient-related factors (e.g., age, general condition, and comorbidities) and treatment modality are unarguably essential in predicting the prognosis of acute myeloid leukemia (AML). Nevertheless, tumor-related genetic factors still have a major impact on prognosis [1,2]. The risk classification of AML based on 2017 European LeukemiaNet (ELN) recommendations (Table 1) has been widely adapted in general practice [3]. The original intention of the risk classification was to refine prognostication and improve decision-making for AML patients at diagnosis. The ELN guideline provides a three-group classification (favorable, intermediate, adverse) according to expected overall survival (OS) and relapse rates. Notably, internal tandem duplication (ITD) of the FMS-like tyrosine kinase 3 (*FLT3*) gene is described in association with nucleophosmin 1 (*NPM1*) mutation status, with only a high allelic ratio (>0.5) of *FLT3*-ITD considered a poor prognostic factor. Mutations in the runt-related transcription factor 1 (*RUNX1*) and additional sex combs such as the 1 (*ASXL1*) and tumor antigen p53 (*TP53*) genes were recently added as adverse genetic abnormalities in the 2017 edition. Many of the genetic factors described in this recommendation are closely related to one another. For example, the *NPM1* mutation (*NPM1*mt) is considered to be a favorable factor only if *FLT3*-ITD is absent or has a low allelic ratio (<0.5) [4]. Concurrent mutations of *ASXL1* and *RUNX1* exert poor prognosis, particularly in younger adults [5]. In core-binding factor (CBF) AML, the co-existing KIT proto-oncogene receptor tyrosine kinase (*KIT*) gene mutation is associated with worse outcome [6]; although, the ELN classification does not include *KIT* mutation because the negative effect can be negated if a three-log or deeper reduction in minimal residual disease is achieved after induction therapy [7]. Although genetic relationships in AML can be overwhelmingly complex and multifactorial, the ELN classification is a pivotal clinical tool in predicting the outcomes of patients with AML.

AML with adverse genetic factors, as defined in the ELN classification, uniformly shows resistance to standard chemotherapy and is associated with poor prognosis. Here, we focus on the biological background and real-world etiology of the adverse genetic factors and then describe a strategy to overcome the clinical disadvantages in terms of targeting pivotal molecular mechanisms. In a comprehensive survey of the available preclinical literature, we searched the literature using the key words “ELN”, “FLT3”, “NPM1”, “DEK-NUP214”, “KMT2A”, “BCR-ABL1”, “GATA2”, “MECOM”, “EVI1”, “RUNX1”, “ASXL1”, “TP53”, “complex karyotype”, “menin”, and “anti-cd47 antibody” with the necessary key word “AML”. A systematic literature search was performed using PubMed and the Tip Medical Database, identifying a total of 1839 records including 92 controlled trials, 90 systematic reviews, and 1228 ongoing clinical trials. We chose 169 records according to the theme of this review article.

## 2. Biology of Adverse Genetic Abnormalities

### 2.1. DEK-NUP214 Fusion

Normal AML with the fusion of the DEK proto-oncogene (DEK) with the nucleoporin 214 (*NUP214*) gene or t(6;9)(p23;q34) is a distinct WHO classification category of AML [8] and is associated with poor prognosis [9]. t(6;9)(p23;q34) is a rare chromosomal abnormality (0.5–4% of AML) and is primarily found in AML with maturation (M2) or acute myelomonocytic leukemia (M4) [10]. DEK protein is a ubiquitous chromatin protein that serves as an architectural protein at promoter or enhancer sites of a subset of human genes [11]. The upregulation of the *DEK* gene has been clinically noted in many human cancers [12]. The NUP214 protein, also known as CAN, is a component of the nuclear pore complex and plays an important role in the nucleocytoplasmic transport of macromolecules [13,14]. Xenografted mice with human hematopoietic progenitors with t(6;9)(p22;q34) developed AML with a myelomonocytic immunophenotype [15]. In the leukemic cell, the gene expression of the homeobox (*HOX*) family genes, such as homeobox protein A9 (*HOXA9*), *HOXA10*, *HOXB3*, *HOXB4*, and *PBX* homeobox 3 (*PBX3*), is highly upregulated. Other studies have suggested that *DEK*-*NUP214* is associated with the upregulation of mechanistic targets of rapamycin (mTOR) [16] and the activation of the signal transducer and activator of transcription 5 (STAT5) [17].

### 2.2. KMT2A (MLL) Rearrangement

The lysine methyltransferase 2A (*KMT2A*) gene, previously known as mixed lineage leukemia 1 (*MLL*), is a proto-oncogene that encodes a histone methyltransferase essential in hematopoiesis [18]. A multiprotein complex containing KMT2A mediates the methylation of the fourth lysine residue in histone H3 (H3K4), as well as the acetylation of the sixteenth lysine residue in histone H4 (H4K16) [19], which is required for the upregulation of *HOXA9* gene transcription [18]. The *KMT2A* gene forms oncogenic fusions with a variety of partners, including mixed-lineage leukemia translocated to chromosome 3 (*MLLT3*), AF4/FMR2 family member 1 (*AFF1*), and the afadin adherens junction formation factor (*AFDN*) [20]. Rearrangements of *KMT2A* are commonly found in AML (approximately 5–10%) and are associated with poor prognosis [21,22]. Recent studies have suggested that oncogenic KMT2A fusion proteins interact with menin and the disruptor of telomeric silencing 1-like (DOT1L), which is required for initiating leukemogenesis [23,24] through the upregulation of specific genes such as *HOXA9* and myeloid ecotropic viral insertion site 1 (*MEIS1*) [25,26]. Menin, a product of the multiple endocrine neoplasia type 1 (*MEN1*) tumor suppressor gene, is critically involved in leukemogenic mechanisms by linking KMT2A protein with the lens epithelium-derived growth factor (LEDGF) on target genes [27]. 

### 2.3. BCR-ABL1 Fusion

ABL proto-oncogene 1 (ABL1) is a non-receptor tyrosine kinase involved in cell growth and survival. Normal ABL is principally located in the nucleus and regulates DNA damage-induced apoptosis [28,29], as well as cytoskeletal remodeling, receptor endocytosis, and autophagy. Several breakpoints have been recognized in pathogenic translocations between chromosome 9q34, the *ABL* gene location, and chromosome 22q11, the breakpoint cluster region protein (*BCR*) gene location [30]. The large molecular weight *BCR-ABL1*-encoded fusion protein (210 kDa; major BCR-ABL1) is associated with chronic myeloid leukemia (CML), while the small molecular weight fusion protein (185–190 kDa; minor BCR-ABL1) is associated with acute lymphoblastic leukemia (ALL). The oncogenic BCR-ABL1 fusion protein is located in the cytoplasm and phosphorylates a variety of proteins [31]. Major BCR-ABL1 is particularly associated with phosphatidylinositol 3-kinase (PI3K), which is required for the *BCR-ABL1*-dependent proliferation of CML cells [32]. Major BCR-ABL1, but not normal ABL, binds to the adaptor molecule CRK-like protein (CRKL) [33], which transduces signals to the oncogene product E3 ubiquitin-protein ligase CBL (CBL) [34]. Major BCR-ABL1 is also related to the constitutive activation of the Janus kinase (JAK)/STAT5 pathway [35] and the indirect activation of RAS proto-oncogene (RAS) proteins [36]. Recent studies have revealed that the BCR-ABL1 fusion protein is a client of the molecular chaperone heat shock protein 90 (HSP90) and the inhibition of HSP90 protein induces the degradation of the pathogenic fusion protein in *BCR-ABL1*-positive cell lines [37,38].

Normal AML with the *BCR-ABL1* fusion gene is a rare entity, accounting for approximately 1% of all AML subtypes [39], and is often challenging to distinguish from the first presentation of CML in blastic crisis [40,41]. While the majority of CMLs uniformly rely on major BCR-ABL1, both major and minor BCL-ABL1 can be equivalently found in *BCR-ABL1*-positive AML. A summary of published *BCR-ABL1*-positive AML cases showed that slightly less than 40% of cases belonged to AML not otherwise specified (AML-NOS) and approximately a quarter of the cases were found to be AML with recurrent genetic aberrations such as CBF-AML and AML with an *NPM1* mutation [41]. Notably, when the ELN criteria were applied independently of *BCR-ABL1* status, more than three quarters of the *BCR-ABL1*-positive AML cases were classified as belonging to the non-favorable risk group [41]. Given such unfavorable genetic backgrounds, the prognostic impact of *BCR-ABL1* itself has not been confirmed.

### 2.4. Haploinsufficiency of GATA2 and Mis-Expression of MECOM

The chromosomal translocation and inversion between 3q21 and 3q26 (3q-rearrangement) are well-known recurrent genetic abnormalities in myelodysplastic syndrome (MDS) and AML [42]. The *MDS1* and *EVI1* complex locus (*MECOM*) gene, formerly known as ecotropic virus integration site 1 protein homolog (*EVI1*), is located in 3q26 and encodes the transcription regulator protein EVI1 which binds to the promoter region of target genes. EVI1 protein regulates apoptosis through Jun N-terminal kinase (JNK) and tumor growth factor (TGF)-beta signaling [43,44]. Increased expression of the *MECOM* gene has been observed in AML with 3q-rearrangement [42]. Katayama and colleagues demonstrated that the inappropriate expression of the *MECOM* gene in AML with 3q-rearrangement was induced by the recruitment of a *GATA2*-distal hematopoietic enhancer (G2DHE) to the proximity of the *MECOM* gene, which subsequently disturbed the function of an allele of the endothelial transcription factor GATA-2 (*GATA2*) gene [45]. The GATA2 protein was originally identified as a transcription activator that regulates endothelin-1 gene expression in endothelial cells and is also highly expressed in normal hematopoietic progenitor cells [46]. The germ line heterozygous mutation of *GATA2* is known to be associated with bone marrow failure, immunodeficiency, lymphedema, and organ dysfunction as well as MDS and AML [47,48,49]. Considering the report from Dr. Katayama described above, the haploinsufficiency of the *GATA2* gene, which is caused by the mis-expression of EVI1 protein, is expected to play a key role in the leukemogenesis of AML with 3q-rearrangement.

3q-rearrangements, such as t(3;3)(q21;q26) or inv(3;3)(q21;q26), are rare cytogenetic abnormalities that are found in approximately 4% of AML [50] and are associated with poor prognosis [50,51,52]. An analysis of a database of 288 patients with 3q-rearranged AML indicated that the overexpression of EVI1 was observed in 95% of patients with t(3;3)/inv(3;3) and the hazard ratio of death increased by 40% [50]. Interestingly, 3q-rearrangement was frequently associated with monosomy 7 and *NRAS* mutation.

### 2.5. Complex Karyotype

In the ELN classification, a complex karyotype is defined as three or more chromosomal abnormalities in the absence of the WHO-designated recurring translocations or inversions, such as t(8;21), inv(16) or t(16;16), t(9;11), t(v;11)(v;q23.3), t(6;9), inv(3), or t(3;3), whereas the UK National Cancer Research Institute Adult Leukaemia Working Group requires four or more chromosomal abnormalities as an adverse risk factor [9]. Complex karyotypes are found in approximately 10% of all AML and are more prevalent (up to 20%) in the elderly [53]. An analysis of a database of 417 AML patients with complex karyotypes showed that three or more chromosomal abnormalities in the absence of strong factors, such as a hyperdiploid karyotype, CBF-AML, and unique adverse-risk aberrations, were associated with reduced OS compared with patients with a normal karyotype [54].

### 2.6. FLT3-ITD with Wild-Type NPM1

FLT3 is a receptor tyrosine kinase that is expressed in normal hematopoietic progenitor cells and regulates proliferation and survival. FLT3 dimerizes upon binding with FLT3 ligands that are produced by bone marrow stromal cells, which results in the phosphorylation of the tyrosine residues in the activation loop [55]. Downstream FLT3 signaling involves several pathways. Activated FLT3 promotes the phosphorylation of tyrosine-protein phosphatase non-receptor type 11 (PTPN11), also known as src homology region 2 domain-containing phosphatase 2 (SHP2) and AKT serine/threonine kinase 1 (AKT1) to activate the downstream effector protein mTOR [56,57]. mTOR is a serine/threonine kinase that functions as a part of two distinct signaling complexes—mTOR complex 1 (mTORC1) and 2 (mTORC2)—and regulates the phosphorylation of a wide range of proteins associated with cellular metabolism, growth, and survival [58,59,60]. FLT3 also phosphorylates the major RAS signaling downstream kinases, namely, mitogen-activated protein kinase (MAPK) and extracellular signal-regulated kinases (ERK) [61]. MAPK and ERK, especially the pairs of MAPK1/ERK2 and MAPK3/ERK1, are involved in an essential component of MAP kinase signaling, regulating a variety of transcription factors (e.g., B-cell lymphoma 6 protein (BCL6) [62]), cytoskeletal elements, and regulators of apoptosis (e.g., myeloid cell leukemia 1 (MCL1) [63]). Although wild-type FLT3 does not significantly affect STAT5 signaling, mutant FLT3 constitutively activates these effector proteins, resulting in the promotion of proliferation and survival [61]. 

Nucleophosmin 1 (NPM1) protein is involved in a variety of cellular mechanisms including ribosome nuclear export [64], the chaperoning of histones [65], centrosome duplication in concert with breast cancer susceptibility gene 2 (BRCA2) [66], promoting proliferation by antagonizing activating transcription factor 5 (ATF5)-induced G2/M cell cycle blockade [67], the regulation of tumor protein p53 (TP53) and the ADP-ribosylation factor (ARF) [68,69], and the enhancement of myc proto-oncogene (*MYC*) transcription [70]. Overall, NPM1 functions in supporting cell proliferation and survival. Wild-type NPM1 protein is predominantly located in the nucleus and shuttles between the nucleus and the cytoplasm. However, because *NPM1* mutations are concentrated in the nuclear export signal (NES) domain, which enables the export of the protein from the nucleus to the cytoplasm, mutant NPM1 protein inappropriately accumulates in the cytoplasm [71]. Indeed, *NPM1* mutation leads to the cytoplasmic localization of several nuclear proteins associated with DNA repair and apoptosis [72], as well as transcription factors such as CCCTC-binding factor (CTCF) and PU.1 [73,74]. The upregulation of *HOX* genes has been reported in AML with NPM1 mutation [75]. Recent studies have suggested that the interaction between menin and wild-type KMT2A plays a pivotal role in *NPM1*-mutated AML through upregulating *HOXA*, *HOXB,* and *MEIS1* [76,77,78].

*FLT3*-ITD—a common pattern of activating mutation that leads to the extension of the juxta-membrane domain of FLT3 tyrosine kinase—and *NPM1* mutation have opposite prognostic impacts in AML: *NPM1* mutation can be a favorable factor only in the absence of *FLT3*-ITD, and the presence of *NPM1* mutation attenuates the adverse prognostic impact of *FLT3*-ITD [79]. Although the current ELN risk classification does not distinguish wild-type *FLT3* from *FLT3*-ITD with a low allelic ratio (*FLT3*-ITD^low^), some reports have suggested that concurrent *NPM1* mutation and *FLT3*-ITD^low^ is not related to favorable outcomes [80]. In terms of influence on treatment, a retrospective analysis of the RATIFY trial—a phase 3 trial that demonstrated the efficacy of adding midostaurin to standard chemotherapy in patients with *FLT3*-mutated AML—indicated that the first-generation FLT3 inhibitor was equally effective for all ELN risk groups regardless of *NPM1* mutation status [81]. 

### 2.7. RUNX1 Loss-of-Function Mutation

The runt-related transcription factor 1 (RUNX1), previously known as acute myeloid leukemia 1 (AML1) or core-binding factor subunit alpha-2 (CBFA2), protein forms a heterodimeric complex, called CBF, with the core-binding factor B (CBFB) protein. The RUNX1 protein recognizes certain DNA sequence patterns, such as 5′-TGTGGT-3′, in the target domain and CBFB allosterically enhances RUNX1 function. The CBF complex interacts with promoters and enhancers of a variety of genes and plays an essential role in normal hematopoiesis [82]. A recent study has suggested that *RUNX1* gene silencing results in the upregulation of TP53, which stabilizes the RUNX1 protein via the enhanced expression of the *CBFB* gene [83]. 

The pathogenic fusion of RUNX1 with its partner transcriptional co-repressor 1 (*RUNX1T1*), previously known as eight twenty one (*ETO*), plays a critical role in the leukemogenesis of CBF-AML via the aberrant transcription factors that contain oncogenic RUNX1-RUNX1T1 or AML1-ETO, fusion proteins [84]. The presence of the *RUNX1-RUNX1T1* fusion gene, or t(8;21)(q22;q22.1) translocation, is generally considered a favorable prognostic factor in AML [85]; however, a database analysis revealed that *RUNX1* mutation, accounting for 10% of newly diagnosed AML, was independently associated with poor prognosis and co-mutations with epigenetic modifiers (e.g., additional sex combs-like 1 (*ASXL1*)) and/or spliceosome-related genes (e.g., serine/arginine-rich splicing factor 2 (*SRSF2*) and plant homeodomain-like finger (*PHF6*)) predicted notably worse outcomes [86]. A patient-derived xenograft (PDX) mouse model experiment using a human chronic myelomonocytic leukemia (CMML) cell line demonstrated that concurrent mutations of *RUNX1* and *ASXL1* promoted cell renewal in bone marrow and leukemic transformation, which was induced by the augmented production of hypoxia-inducible factor 1 subunit alpha (HIF1-α) and the consequent upregulation of the inhibitor of DNA binding 1 (ID1) and AKT1 signaling [87]. ID1 is a negative transcriptional regulator and is associated with the immortalization of hematopoietic progenitor cells [88]. On the other hand, a mouse model experiment suggested that Runx1 protein cooperates with Pu.1, and the loss of Runx1 results in the decreased ability of Pu.1 to promote the differentiation of the macrophage lineage [89].

### 2.8. ASXL1 Loss-of-Function Mutation

ASXL1 protein recognizes adenine N6 methylation in DNA (6 mA) and is a component of the Polycomb repressive deubiquitinase (PR-DUB) complex. PR-DUB removes monoubiquitylation of the 119th lysine residue in histone H2A (H2A-K119Ub) and is degraded following 6 mA formation [90]. H2A-K119Ub is a well-known histone marker related to repressed transcription [91], whereas Polycomb group (PcG) proteins play a critical role in maintaining appropriate gene repression by forming Polycomb repressive complexes (PRC). Contrary to PR-DUB, PRC1 is responsible for the formation of H2A-K119Ub. However, recent studies have suggested that PR-DUB does not simply counteract PRC1, but rather functions in concert with PRC1 [92,93,94]. A mouse model experiment demonstrated that *Asxl1* mutation is accompanied by aberrant PRC1-mediated histone modification, resulting in impaired hematopoiesis similar to low-risk MDS [92]. Overall, the *ASXL1* gene acts as a tumor suppressor, at least in terms of hematopoietic regulation.

Nonsense mutation or the frame-shift of the *ASXL1* gene, mainly in exon 11 or 12, is found in approximately 10% of AML and is more prevalent in MDS (10–25%) and CMML (40–50%). *ASXL1* mutation is known to be mutually exclusive with DNA-methyltransferase 3 alpha (*DNMT3A*), *NPM1*, and splicing factor 3b subunit 1 (*SF3B1*) mutations [95]. In mouse model studies, *Asxl1* along with Tet methylcytosine dioxygenase 2 (*Tet2*) mutation initiated MDS-like hematopoietic impairment [92,96], and concurrent oncogenic mutations such as *Kras* and *Nf1* accelerated the development of myeloid leukemia [97]. *ASXL1* mutation is known as an independent adverse prognostic factor of AML [98] and concurrent mutation with *RUNX1* is particularly associated with worse prognosis [86].

### 2.9. TP53 Loss-of-Function Mutation

TP53 protein acts as a major tumor suppressor in many tumor types. TP53 regulates cell division by activating inhibitors of cyclin-dependent kinases (e.g., p21) [99] and induces apoptosis via the stimulation of BCL-2-associated X (BAX) [100] as well as the repression of B-cell/CLL lymphoma 2 (BCL-2) [101]. The pro-apoptotic BAX protein, as well as the BCL-2 homologous antagonist/killer (BAK) protein, is expressed in the outer mitochondrial membrane and is usually deactivated by binding with BCL-2, which also captures cytoplasmic pro-apoptotic BCL-2 homology 3 (BH3)-only proteins. Under apoptotic signaling, free cytoplasmic BH3-only proteins activate the BAX protein which results in mitochondrial outer membrane permeabilization (MOMP) and the release of cytochrome c into the cytoplasm, which consequently leads to apoptosis [102,103]. TP53 protein is physiologically neutralized by mouse double minute 2 homolog (MDM2). The tumor suppressor protein p14^ARF^, an isoform product of the cyclin-dependent kinase inhibitor 2A (*CDKN2A*) gene that is upregulated in response to cell proliferation signals, inhibits MDM2 and thereby activates TP53 [104]. 

*TP53* mutation is found in less than 10% of normal AML and 30–40% in secondary AML [105]. *TP53* mutation in AML is generally associated with complex karyotypes and universally poor prognosis [106]. TP53-mutated leukemic cells gain enhanced self-renewal capacity and a competitive growth advantage, subsequently accumulating additional mutations, including *DNMT3A*, *TET2*, and *ASXL1* [107]. Interestingly, *TP53*-mutated secondary AML and MDS are associated with the increased expression of programmed death ligand 1 (PD-L1), a major immune checkpoint molecule that leads to the anergy of effector T cells on hematopoietic stem cells and decreased numbers of cytotoxic/helper T cells in the bone marrow [108]. 

The pro-leukemic mechanisms of adverse genetic factors are schematically summarized in Figure 1. The upregulation of *HOXA9* is a common phenomenon in AML with *KMT2A* rearrangement, *NPM1* mutation, and *DEX*-*NUP214* fusion. The aberrant tyrosine kinase activity of *FLT3*-ITD and *BCR*-*ABL1* fusion affects three major signaling pathways: the AKT-mTOR, MAPK-ERK, and STAT5 pathways. Suppressed apoptosis is caused by *TP53* mutation and *MECOM* mis-expression.

## 3. Real-World Etiology of Adverse Genetic Abnormalities

### 3.1. Ad Hoc Analysis of the Japan Adult Leukemia Study Group (JALSG) AML201 Study

JALSG AML201 was a Japanese multi-center phase 3 randomized study evaluating the equality of standard-dose idarubicin and high-dose daunorubicin as induction therapies, as well as high-dose cytarabine and standard-dose chemotherapies as consolidation therapies in previously untreated AML [109,110]. The ad hoc analysis of the AML201 study focused on 197 patients in which comprehensive genetic data were available [85]. Favorable, intermediate, and adverse risk groups accounted for 28%, 60%, and 12% of the patient population, respectively. Among adverse genetic factors found in this study, complex karyotype was the most prevalent (8.1%), followed by *RUNX1* mutation (5.1%) and the rearrangement of the *KMT2A* gene (3.0%). Five-year OS was 59.1% in the favorable risk group, 32.6% in the intermediate risk group, and 22.6% in the adverse risk group.

### 3.2. Hematologic Malignancy (HM)-SCREEN JAPAN 01 Study

This section is composed of a detailed analysis of published data from the Japanese genomic sequencing study HM-SCREEN-Japan 01 [111,112]. Although this study included only a small number of AML patients, it can serve to highlight the characteristics of patients in which conventional treatment was not successful or recommended.

#### 3.2.1. Study Design

HM-SCREEN-JAPAN 01 was a multicenter genomic profiling study in which next-generation sequencing by FoundationOne Heme^®^ was performed for patients with AML. The inclusion criteria were patients with refractory/relapsed AML (R/R) or with previously untreated AML who were ineligible for standard therapy (ND unfit). The available specimens were paraffin-embedded bone marrow clots. The submission of archival specimens was allowed. After submission, annotated genomic reports were returned to the participants. Patients with the *FLT3* mutation were allowed to submit specimens multiple times at different opportunities. Clinical information was gathered, including age, sex, treatment modality, response, stem cell transplantation status, and survival information. 

#### 3.2.2. ELN Classification

Using conventional cytogenetic tests and next-generation sequencing (NGS), the patients were classified as belonging to non-adverse (favorable or intermediate) or adverse risk groups according to the 2017 ELN recommendation. In Japan, commercially available cytogenetic tests include G-band karyotyping, polymerase chain reaction (PCR) for recurrent rearrangements, *FLT3*-ITD, and *NPM1* mutation. However, these PCR data were not available in the HM-SCREEN-JAPAN 01 study. To supplement the lack of data on *FLT3*-ITD and *NPM1*, “classification by conventional cytogenetic tests” was defined as that with G-band karyotyping plus *FLT3-ITD* and *NPM1* information obtained from NGS analysis. Due to the lack of *FLT3*-ITD allelic ratio assessment, we considered all patients with *FLT3*-ITD and *NPM1* wild type as belonging to the adverse risk group. NGS was performed on bone marrow samples either at the time of diagnosis or relapse.

#### 3.2.3. Conventional Versus NGS-Based Classification

The clinical and cytogenetic characteristics are summarized in Appendix A. One hundred eighty-two patients were enrolled and fourteen patients were excluded due to unavailable data on G-band karyotyping. Thus, 168 patients were subsequently analyzed, of which 66 (39.3%) and 102 (60.7%) had newly diagnosed and relapsed/refractory AML, respectively. Regarding disease type, 117 (69.6%) patients were diagnosed with normal AML while 41 (24.4%) had AML with myelodysplasia-related changes and 10 (6.0%) had therapy-related myeloid neoplasms. The median age was 63 years (range: 20–91). First, the conventional cytogenetic tests classified 105 (62.5%) patients into the non-adverse risk group and 63 (37.5%) into the adverse risk group. Among the patients in the adverse risk group, the most frequently observed cytogenetic aberration was complex karyotype (25 patients, 39.6%), followed by *FLT3*-ITD in the absence of mutated *NPM1* (15 patients, 23.8%), and *KMT2A* gene rearrangement (7 patients, 11.1%). Next, we performed NGS-based classification. The timing of NGS analysis was at diagnosis in approximately half of the patients (48.8%). Contrary to the conventional cytogenetic tests, NGS detected an additional 88 adverse gene aberrations, resulting in the re-classification of 39 non-adverse risk patients as adverse risk. Consequently, 66 (39.3%) patients were grouped as non-adverse risk and 102 (60.7%) as adverse risk (Figure 2). Of the re-classified 39 patients, 19 (48.7%), 14 (35.9%), 6 (15.4%), 4 (10.3%), and 4 (10.3%) harbored mutations of *ASXL1*, *RUNX1*, *GATA2*, *TP53* and *KMT2A* gene rearrangement, respectively. Approximately half of these gene aberrations (51.3%) were detected in the samples at diagnosis. Notably, one re-classified patient harbored the sole TP53 mutation in the specimen at diagnosis without adverse chromosomal abnormalities. Co-mutation was most frequently observed with *ASXL1* and *RUNX1* in 5/19 (26.3%) patients. Among the 63 patients in the adverse risk group who were not re-classified, NGS detected 49 adverse gene aberrations, of which *TP53* mutation was the most frequently found in 32 (50.8%) patients. 

Contrary to the re-classified patients, *ASXL1* (4, 6.3%) and *RUNX1* (7, 11.1%) mutations were found less frequently in the 63 patients in the adverse risk group based on the conventional cytogenetic tests. Moreover, the majority of *TP53* mutations (72%) co-occurred with a complex karyotype or a specific aneuploidy involving chromosomes 5 and 7.

#### 3.2.4. Clinical Outcome

After enrollment, many of the patients were treated with non-intensive treatment (64.9%) or best supportive care (9.5%), while 24.4% were treated with intensive chemotherapy. Fifty-nine patients (35.1%) underwent subsequent allogeneic hematopoietic stem cell transplantation (HSCT). When classified based only on the conventional cytogenetic tests, 2-year OS and 2-year progression free survival (PFS) in the non-adverse group was 61.6% (95% confidence interval (CI); 49.3–71.8%) and 46.1% (95% CI; 34.6–56.9%), respectively, while significantly worse outcomes were observed in the adverse group, with a 2-year OS of 39.7% (95% CI; 25.4–53.6%, *p* = 0.01) and a 2-year PFS of 32.3% (95% CI; 19.9–45.4%, *p* = 0.02). With NGS analysis, prognosis was similarly discriminated between the non-adverse and adverse groups; a 2-year OS of 68.1% (95% CI; 52.4–79.6%) and a 2-year PFS of 47.9% (95% CI; 33.1–61.3%) in the non-adverse group versus significantly worse outcome in the adverse group, with a 2-year OS of 43.5% (95% CI; 31.7–54.7%, *p* = 0.01) and a 2-year PFS of 36.5% (95% CI; 26.1–47.0%, *p* = 0.02). Of note, patients with mutated *TP53* had particularly unfavorable outcomes with a 2-year OS of 24.1% (95% CI; 10.4–40.9%) and a 2-year PFS of 17.1% (95% CI; 6.2–32.6%). Patients with mutated *TP53* who received HSCT exhibited significantly improved OS compared to those who did not (median OS; 24.5 months, 95% CI; 7.4–38.8, versus 6.8 months, 95% CI; 4.2–8.8, *p* < 0.001).

In summary, the HM-SCREEN-Japan01 study suggested that adverse genetic factors are accumulated in patients with relapsed/refractory AML or those who are ineligible for intensive chemotherapy. Among these genetic abnormalities, *TP53* mutation was associated with especially poor outcome. Long-term survival would be expected in patients who underwent HSCT, accounting for only a third of participants of this study (Appendix A). Molecular-targeting agents specific for adverse genetic factors may be the key to improving the clinical outcomes of patients who are not eligible to HSCT and/or intensive treatment.

## 4. How to Deal with Specific Adverse Genetic Factors

### 4.1. FLT3 Inhibitors

Potent FLT3-specific inhibitors such as midostaurin, gilteritinib, and quizartinib are currently available in practice. In the randomized phase 3 RATIFY trail, standard induction therapy (daunorubicin and cytarabine (DNR/AraC), followed by consolidation with high-dose cytarabine) in combination with midostaurin showed longer OS and event-free intervals than chemotherapy alone in patients newly diagnosed with FLT3-mutated AML [113]. The combination of midostaurin and standard therapy followed by midostaurin maintenance also showed better outcomes compared with historical controls (hazard ratio 0.58 in event-free survival) [114]. Gilteritinib monotherapy prolonged survival (9.3 months vs. 5.6 months) compared with conventional salvage therapy (e.g., mitoxantrone, etoposide, and cytarabine (MEC); fludarabine, cytarabine, granulocyte colony-stimulating factor, and idarubicin (FLAG-IDA); low-dose cytarabine (LDAC); and azacitidine) in patients with relapsed/refractory *FLT3*-mutated AML in the randomized phase 3 ADMIRAL trial [115]. Similarly, quizartinib monotherapy improved OS (hazard ratio 0.76) compared with conventional salvage therapy (e.g., LDAC, MEC, and FLAG-IDA) in the phase 3 QuANTUM-R trial [116]. The novel second-generation FLT3 inhibitor crenolanib has shown possible benefits in combination with conventional chemotherapy in both first-line (anthracyclines/AraC with high-dose AraC) and salvage (high-dose AraC/mitoxantrone) treatment [117,118,119]. In addition to salvage therapy, FLT3 inhibitors have also been evaluated in upfront combination therapy. In a phase 1 study, gilteritinib in combination with a standard 3 + 7 regimen (idarubicin and cytarabine (IDA/AraC)) showed high response rates (89%) with acceptable toxicity in patients with *FLT3*-mutated AML [120]. A phase 2 randomized trial comparing gilteritinib versus midostaurin in combination with standard chemotherapy (DNR/AraC) is now ongoing [121]. Front-line combination with quizartinib has also been evaluated in the double-blind phase 3 QuANTUM-First trial [122], and its press release announced superior overall survival compared with standard therapy (DNR/AraC or IDA/AraC) alone in patients with newly diagnosed *FLT3*-mutated AML [123].

### 4.2. Inhibition of the AKT, MAPK, and STAT Pathways

The PI3K-AKT-mTOR, RAS-MAPK-ERK, and JAK2-STAT5 pathways are major downstream signals of a variety of tyrosine kinases, including FLT3 and ABL1. The amplification of the PI3K-AKT-mTOR pathway is seen in at least 60% of patents with AML [124]. In addition to the *FLT3*-ITD and *BCR-ABL1* fusion described above, the amplification of AKT signaling can be caused by KIT activation [125], RAS mutations [126], and the overexpression of PI3K [127]. Although mTOR inhibitors as monotherapy showed only a limited efficacy on AML [128,129], dual PI3K/mTOR inhibitors (e.g., NVP-BEZ235, NVPBGT226, and PI-103) induced the cell cycle arrest and apoptosis of leukemic cell lines in preclinical studies [130,131,132]. 

Increased levels of phosphorylated JAK2 and STAT5 were noted in AML patient’s bone marrow samples [133,134]. Activating the mutation of *FLT3* and *JAK2* (e.g., V617F), as well as the functional loss of the suppressor of cytokine signaling-1 (*SOCS1*) leads to the constitutive activation of JAK2-STAT5 signaling in AML [135]. To date, three JAK inhibitors—ruxolitinib, lestaurtinib, and pacritinib—have been evaluated in clinical studies for AML and/or high-risk myeloproliferative neoplasms (MPNs). Although ruxolitinib failed to demonstrate an appropriate clinical benefit as monotherapy in relapsed/refractory AML [136], the combination therapy of ruxolitinib and decitabine showed a response rate of 61% with tolerable toxicity in patients with high-risk MPN [137]. The sequential administration of lestaurtinib after chemotherapy failed to show a clinical benefit in patients with *FLT3*-mutated AML in their first relapse in a phase 3 trial [138]. Pacritinib was administered in a small number of patients with relapsed/refractory AML and showed a clinical benefit rate of 43% [139]. 

Mutations in genes that regulate the RAS-MAPK-ERK pathway, such as *NRAS*, *KRAS*, *PTPN11*, *NF1*, and *KIT* are commonly found in patients with AML, and RAS-MAPK-ERK signaling is often highly activated [140]. However, inhibitors of mitogen-activated protein kinase kinase (MEK), a key protein of the RAS-MAPK-ERK pathway, have failed to show meaningful anti-leukemic activity in several clinical studies of either RAS-mutant and RAS wild-type AML [141,142,143]. RAF proto-oncogene (RAF) kinases also play a key role in this pathway and three isoforms (A-RAF, B-RAF, and C-RAF) have been identified to date. Recently, a novel pan-RAF inhibitor (LY3009120) has demonstrated promising anti-tumor activity in solid tumors and multiple myeloma [144,145]. Tambe and colleagues demonstrated that pan-RAF inhibition induced leukemic cell death in 29% of samples from AML patients [146]. Interestingly, pan-RAF inhibition accompanied the downregulation of MCL1 protein, a negative regulator of apoptosis, and showed synergistic anti-leukemic activity in combination with a BCL-2 inhibitor.

### 4.3. Menin-KMT2A Inhibitors

As mentioned in the previous section, AML with *KMT2A* rearrangement (*KMT2A*r) is pathologically characterized by upregulated *HOXA9* and *MEIS1* genes, which are dependent on the interaction of oncogenic KMT2A fusion proteins with other complex-forming proteins such as LEDGF, DOT1L, and menin. Strategies for disrupting the connection between KMT2A protein and menin have recently been investigated. A preclinical study in which PDX mice with *KMT2A*r were treated with the orally bioavailable menin-KMT2A inhibitor VTP50469 demonstrated that *KMT2A*-target genes such as *MEIS1* were uniformly suppressed in the bone marrow [147]. Another menin-KMT2A inhibitor, MI-3454, also showed significant anti-leukemic effects along with the downregulation of the *MEIS1* gene in a PDX model experiment [148]. Fiskus and colleagues also demonstrated that the menin-KMT2A inhibitor SNDX-50469 induced differentiation and reduced the viability of the leukemic cell lines with *KMT2A*r or *NPM*mt, which was accompanied by the attenuation of BCL-2 and cyclin-dependent kinase 6 (CDK6) levels. Interestingly, the concurrent administration of SNDX-50469 with a BCL-2 inhibitor or a CDK6 inhibitor led to synergistic lethality [149]. Combination therapy with memin-KMT2A inhibitors and FLT3 inhibitors represents a promising strategy for AML with *KMT2A*-r or *NPM1*mt and concurrent *FLT3* mutation (*KMT2A*r/*NPM1*mt-*FLT3*mt). Dzama and colleagues showed the synergistic inhibition of the proliferation and enhancement of apoptosis in leukemic cell lines with *KMT2A*r/*NPM1*mt-*FLT3*mt when treated with FLT3 inhibitors and menin-KMT2A inhibitors [150]. Notably, the first-in-human phase 1 AUGMENT101 study of the menin-KMT2A inhibitor SNDX-5613 as monotherapy showed promising results in patients with relapsed/refractory AML carrying *KMT2A* rearrangements or *NPM1* mutations. Among the 59 patients recruited, clinical responses were observed in 28 patients (47%), including 8 patients (14%) with complete remission (CR) for more than 6 months [151]. 

### 4.4. TP53 Stabilizers

A number of small molecules that reactivate mutant TP53 have been identified. Bykov and colleagues demonstrated that one such molecule (PRIMA-1) and its methylated form induced apoptosis in human tumor cells with *TP53* mutation [152,153]. APR-246 (eprenetapopt), an alternative name for methylated PRIMA-1, is a first-in-class agent that thermodynamically stabilizes TP53 protein and shifts the equilibrium towards a functional conformation [154]. In a phase 1b/2 clinical study, combination therapy with APR-246 and azacitidine produced a 71% overall response rate in patients with *TP53*-mutated high-risk MDS or AML [155]. However, the combination strategy did not meet the primary endpoint in a phase 3 clinical trial for patients with TP53-mutated MDS, although the CR rate tended to be superior in the combination group compared to the azacitidine monotherapy group (33.3 vs. 22.4%) [156]. The next-generation TP53 stabilizer APR-548 is now being evaluated in an early-phase clinical trial [157].

Mouse double minute 2 (MDM2) protein is a TP53-specific E3 ubiquitin ligase and functions as a principal cellular antagonist of TP53 [158]. Idasanutlin, a selective MDM2 antagonist, showed a synergistic anti-leukemic effect in combination with a BCL-2 inhibitor in *TP53* wild-type AML cell lines [159]. Another MDM2 inhibitor, RG7112, was evaluated in a phase 1 study for patients with acute and chronic leukemia regardless of *TP53* mutation status [160]. Thirty-three patients with AML received this agent in its maximal tolerated dose and five patients (15.2%), all *TP53* wild-type, achieved complete or partial remission. Thus, MDM2 inhibitors seem appropriate for *TP53* wild-type cases. Novel MDM2 inhibitors (e.g., DS-3032b, AMG-232) in combination with hypomethylating agents (HMAs) are now under evaluation in clinical trials (NCT03634228, NCT03041688) for patients with newly diagnosed and relapsed/refractory AML. 

Grob and colleagues described the genetic background of 230 patients with *TP53*-mutated AML and high-risk MDS, among 2200 participants in the Haemato-Oncology Foundation for Adults in the Netherlands and Swiss Group for Clinical Cancer Research (HOVON-SAKK) clinical trials [161]. The most frequent co-mutations were *DNMT3A*, *TET2*, and *ASXL1* (so-called DTA mutations), accounting for 24.3% of all *TP53*-mutated patients. These DTA mutations are known to result in disturbed epigenetic modulation within the tumor cells and are associated with better survival when treated with HMAs such as azacitidine [162]. In a meta-analysis of genome sequencing studies in MDS treated with HMAs, *TP53* mutation was associated with improved response to HMAs but also linked to worse prognosis [163]. These data suggest that HMA monotherapy may not be sufficient to overcome the adverse impact of *TP53* mutation, but represents a promising component of combination therapy with mutant TP53-specific agents.

### 4.5. Anti-CD47 Antibody

The transmembrane protein CD47, also known as the “don’t-eat-me signal”, is the ligand for signal regulatory protein alpha (SIRP-alpha) on macrophages and dendritic cells, and confers the inhibition of phagocytosis [164]. The binding of CD47 to SIRP-alpha leads to the recruitment of SH2 domain-containing protein tyrosine phosphatase (SHP)1 and SHP2 in the cytoplasm of macrophages, which are negative regulators of cell signaling [165,166]. The expression of CD47 on AML stem cells is associated with poor prognosis [167]. The anti-CD47 antibody magrolimab in combination with azacitidine showed a 57% response rates in patients with previously untreated AML ineligible for intensive therapy in a phase 1b study. Notably, the response rates were slightly better (67%) in *TP53*-mutated AML in this study [168]. A phase 2 study evaluating the combination therapy of magrolimab with various cytotoxic agents for myeloid malignancies is now in progress [169].

Molecular-targeted agents for AML with adverse genetic factors described to date are summarized in Figure 3.

## 5. Conclusions

Treating AML with adverse genetic factors remains challenging, especially in patients who are ineligible for and/or refractory to intensive chemotherapy. In our real-world study, adverse genetic factors were apparently accumulated in patients with relapsed/refractory AML and who were ineligible to intensive therapy, suggesting a relevant need for a new strategy specific for AML with adverse genetic factors. FLT3 inhibitors have been an archetypal example of a successful molecular-targeting strategy for AML with *FLT3* mutations, with the clinical benefit of their front-line use in combination with standard chemotherapy being demonstrated in large-scale clinical trials. However, resistance to FLT3 inhibition is a relevant issue to overcome, and pathologic FLT3 mutations (e.g., *FLT3*-ITD) are not detected in the majority of AML patients. Targeting common pathways essential to adverse genetic factor function, such as HOXA9-related effectors, tyrosine kinase downstream pathways (e.g., AKT-mTOR, MAPK-ERK, and STAT5 pathways), and the loss of TP53 function could be the key to overcoming this issue. Of note, HOXA9 plays a pivotal role in leukemogenesis in AML with *KMT2A* rearrangement, *NPM1* mutation, and *DEK-NUP214* fusion, as well as AML with *NUP98*-involving fusion genes. Given that the interaction of menin and *KMT2A* protein is essential in initiating the oncogenesis of *HOXA9*-driven leukemia, menin-KMT2A inhibitors are promising molecular-targeting agents applicable to a considerable proportion of AML patients. Apart from FLT3 inhibitors, the direct inhibition of the PI3K-AKT-mTOR, RAS-MAPK-ERK, and JAK2-STAT5 pathways has been evaluated in preclinical and clinical studies. JAK2 inhibitors may be beneficial in leukemic states arising from MPNs. Although MEK inhibitors did not demonstrate apparent effectiveness for AML, pan-RAF inhibitors, especially in combination with BCL-2 inhibitors, represent a promising strategy in AML therapy. Specific approaches to AML with *TP53* mutation remain largely investigational. However, a next-generation TP53 stabilizer is under evaluation in an early-phase trial. An anti-CD47 antibody in combination with HMAs/chemotherapy is another potential strategy for dealing with this category of AML, since this agent seemed equally effective for *TP53*-mutant and wild-type AML. Although these targeted agents exhibited only a limited anti-leukemic effect in the form of monotherapy, combination therapy with other agents such as conventional chemotherapy, HMAs, and BCL-2 inhibitors represents a promising new strategy.

## Figures and Tables

**Figure 1 ijms-23-05950-f001:**
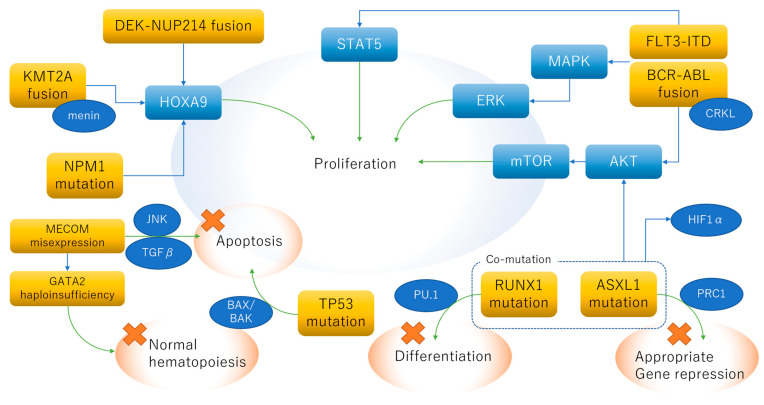
Schematic of pro-leukemic mechanisms of adverse genetic factors. The upregulation of *HOX* genes, especially *HOXA9*, plays a key role in AML with *DEK-NUP219* fusion, *KMT2A* rearrangement, or *NPM1* mutation. FLT3-ITD and BCR-ABL1 fusion proteins are major tyrosine kinases that promote cell proliferation through the activation of the AKT-mTOR, MAPL-ERK, and STAT5 pathways. The concurrent mutation of *ASXL1* and *RUNX1* is associated with the upregulation of AKT and HIF1-α. Decreased p53 function and the overexpression of EVI1 lead to impaired apoptotic mechanisms.

**Figure 2 ijms-23-05950-f002:**
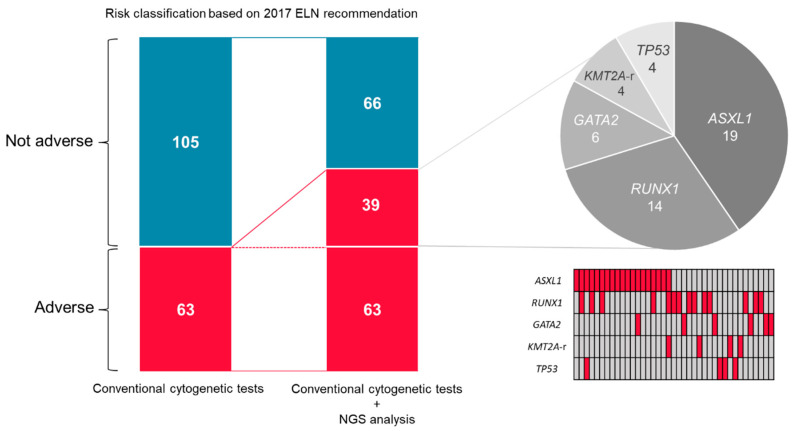
Detailed analysis of the published data of HM-SCREEN-Japan 01 [111,112]. Among 168 patients whose chromosomal status was known, 63 patients (37.5%) were classified as adverse risk according to the ELN2017 criteria by applying conventional cytogenetic tests only. An additional 39 patients (102 total patients (60.7%)) were classified as adverse risk after referring to the NGS results. Adverse genetic factors of the additional 39 cases were *ASXL1* mutation in 19 patients (48.7%), *RUNX1* mutation in 14 patients (35.9%), *GATA2* mutation in 6 patients (15.4%), *KMT2A* rearrangement in 4 patients (10.3%), and *TP53* mutation in 4 patients (10.3%).

**Figure 3 ijms-23-05950-f003:**
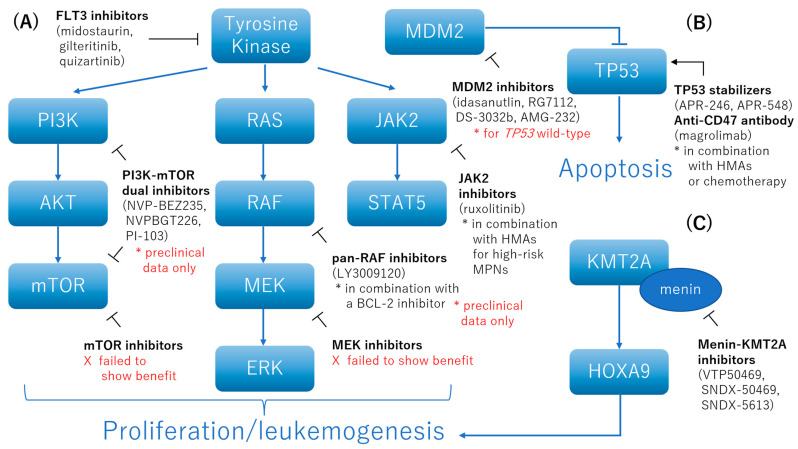
A summary of molecular-targeted agents for AML with adverse genetic factors. (**A**) *FLT3*-ITD and *BCR*-*ABL1* activate the PI3K-AKT-mTOR, RAS-MAPK-ERK, and JAK2-STAT5 pathways to promote leukemic proliferation and leukemogenesis. FLT3 inhibitors (e.g., midostaurin, gilteritinib, and quizartinib) have demonstrated clinical benefit for front-line use in combination with standard induction chemotherapy. Dual PI3K-mTOR inhibitors (e.g., NVP-BEZ235, NVPBGT226, and PI-103) induced cell cycle arrest and apoptosis of leukemic cells in preclinical studies. A pan-RAF inhibitor (LY3009120) led to the downregulation of MCL-1 and showed synergistic anti-leukemic activity in combination with a BCL-2 inhibitor. The combination of ruxolitinib, a JAK2 inhibitor, and decitabine, an HMA, showed high response rates with good tolerability in patients with high-risk MPNs. (**B**) TP53 normally regulates the cell cycle, DNA repair, and apoptosis, and MDM2 counteracts these functions. A first-generation TP53 stabilizer, APR-246, in combination with azacitidine failed to show statistically significant superiority in patients with *TP53*-mutated MDS in a phase 3 study. However, a next-generation TP53 stabilizer, APR-548, is now under evaluation in an early-phase trial. CD47 is an immune-regulatory tumor antigen that inhibits phagocytosis in macrophages. Although the mechanism of CD47 inhibition is dependent on TP53 function, an anti-CD47 antibody, magrolimab, in combination with azacitidine has demonstrated equal effectiveness in patients with AML regardless of *TP53* mutation status. (**C**) AML with *KMT2A* rearrangement, *NPM1* mutation, or *DEK*-*NUP214* fusion genes depends on the upregulation of *HOXA9*, which is initiated by the interaction of menin and KMT2A protein. A menin-KMT2A inhibitor, SNDX-5613, has shown promising results as a monotherapy in patients with relapsed/refractory AML with *KMT2A* rearrangement or *NPM1* mutation. PI3K: phosphatidylinositol 3-kinase, AKT: AKT serine/threonine kinase 1, mTOR: mechanistic targets of rapamycin, MEK: mitogen-activated protein kinase kinase, ERK: extracellular signal-regulated kinases, JAK2: Janus kinase 2, STAT5: signal transducer and activator of transcription 5, MDM2: mouse double minute 2, TP53: tumor protein p53, KMT2A: lysine methyltransferase 2A, HOXA9: homeobox protein A9.

**Table 1 ijms-23-05950-t001:** 2017 ELN risk classification.

Risk Category	Genetic Abnormality
Favorable	t(8;21)(q22;q22.1); *RUNX1-RUNX1T1*
	inv(16)(p13.1q22) or t(16;16)(p13.1;q22); *CBFB-MYH11*
	Mutated *NPM1* without *FLT3*-ITD or with *FLT3*-ITD ^low^
	Biallelic-mutated *CEBPA*
Intermediate	Mutated *NPM1* and *FLT3*-ITD ^high^
	Wild-type *NPM1* without *FLT3*-ITD or with *FLT3*-ITD ^low^ (without adverse-risk genetic lesions)
	t(9;11)(p21.3;q23.3); *MLLT3-KMT2A*
	Cytogenetic abnormalities not classified as favorable or adverse
Adverse	t(6;9)(p23;q34.1); *DEK*-*NUP214*
	t(v;11q23.3); *KMT2A* rearranged
	t(9;22)(q34.1;q11.2); *BCR*-*ABL1*
	inv(3)(q21.3q26.2) or t(3;3)(q21.3;q26.2); *GATA2*,*MECOM(EVI1)*
	−5 or del(5q); −7; −17/abn(17p)
	Complex karyotype, monosomal karyotype
	Wild-type *NPM1* and *FLT3*-ITD ^high^
	Mutated *RUNX1*
	Mutated *ASXL1*
	Mutated *TP53*

## Data Availability

Data sharing not applicable.

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
