# Peer review of "Molecular Classification and Overcoming Therapy Resistance for Acute Myeloid Leukemia with Adverse Genetic Factors"

_ijms, 2022, doi:10.3390/ijms23115950_

Round 1

Reviewer 1 Report

This review manuscript describes strategies for a more efficient treatment of AML. Relevant and specific molecular biology of AML as well as promising therapeutic strategies were described. The manuscript is well written, the figures are clear and informative. There are some minor issues the authors may address to in a revised version of the manuscript: 

Section 4: Please provide a figure with the inhibitor structures and a table summarizing the data presented in this section. Explain the relevance of Akt, MAPK and STAT5 pathways in terms of established and available drugs that inhibit these kinases/targets (apart from FLT3 inhibitors). Can synergy effects be expected when combined with other drugs (p53 activators, etc.), are there any reports about such effects?

Section 4: Azacitidine was used for various combination therapies. Please explain the mode of action and why azacitidine is a suitable combination partner for p53 stabilizers and anti-CD27 antibodies.

Section 4.1: Please specify standard therapy and conventional chemotherapy.

Section 4.3.: Please describe the effects of other p53 activators (e.g., nutlins) on AML.

Section 5, Conclusion: This section is a bit short and rather a summary than a conclusion. Please provide more possible future aspects based on the data shown in this manuscript.

Author Response

We appreciate your precious comments that make this paper more valuable. The reply for your indications and questions are following.

>Section 4: Please provide a figure with the inhibitor structures and a table summarizing the data presented in this section. Explain the relevance of Akt, MAPK and STAT5 pathways in terms of established and available drugs that inhibit these kinases/targets (apart from FLT3 inhibitors). Can synergy effects be expected when combined with other drugs (p53 activators, etc.), are there any reports about such effects?

Reply: We added new paragraphs in section 4, which mentioned about inhibition of AKT-mTOR, JAK2-STAT5, and RAS-RAF-MAP pathways in AML. We also schematically summarized molecular-targeted agents introduced in this section as figure3.

>Section 4: Azacitidine was used for various combination therapies. Please explain the mode of action and why azacitidine is a suitable combination partner for p53 stabilizers and anti-CD47 antibodies.

Reply: We added a new paragraph following description of TP53 stabilizers, which mentioned availability of hypomethylating agents for TP53-mutated AML.

>Section 4.1: Please specify standard therapy and conventional chemotherapy.

Reply: The regimens were specified.

>Section 4.3.: Please describe the effects of other p53 activators (e.g., nutlins) on AML.

We added a new paragraph which mentioned about MDM2 inhibitors.

>Section 5, Conclusion: This section is a bit short and rather a summary than a conclusion. Please provide more possible future aspects based on the data shown in this manuscript.

We wrote a little bit more in conclusion.

Reviewer 2 Report

This manuscript is a mixture of two reviews and the presentation of original data.

The first review, following a short description of the 2017 ELN classification of prognostic factors in acute myeloid leukemia, is a series of small paragraphs describing the structure and function of the anomalies considered in this classification as related to adverse prognosis, presented in the order of the ELN classification.

The second part of the manuscript, strangely dubbed “real-world etiology of adverse genetic abnormalities” is in fact an analysis of two Japanese studies. The first was published more than 10 years ago and is a clinical trial without any clear hint at “real life”. The other one is a genomic project that enrolled relapsed/refractory AML patients or patients ineligible for standard therapy. This is more “real life” since the therapeutic strategies were very heterogeneous. This study has been proposed as an interim analysis in an abstract at ASH 2020 and, more recently, again as an abstract, focusing on the genomic analysis at ASH 2021. Data from these patients are here reanalyzed comparing conventional cytogenetics alone or in combination with high throughput sequencing, demonstrating that 39 more patients were classified as presenting with and adverse profile. Patient outcomes based on these two stratifications are presented. Of note no conclusion nor message is derived from this presentation.

The third part of the manuscript is another review, based on some targeted therapies.

This work clearly lacks a well-defined objective. Moreover, being presented as a review, it should state how the articles were chosen, with which keywords, and the strategy for retaining or discarding the reports retrieved should be presented. Some messages should be clearly derived at least from the clinical study and for the use of the drugs that were chosen to be discussed.

The manuscript is globally reasonably well-written yet should be checked for grammatical errors.

The reference list is curious with the volume number preceding the year of publication, followed by pagination.

Author Response

We appreciate your precious comments that make this paper more valuable. The reply for your indications and questions are following.

>This work clearly lacks a well-defined objective. Moreover, being presented as a review, it should state how the articles were chosen, with which keywords, and the strategy for retaining or discarding the reports retrieved should be presented. Some messages should be clearly derived at least from the clinical study and for the use of the drugs that were chosen to be discussed.

We added method of searching literature in the introduction section. A summary figure of molecular-targeted therapy introduced in this paper was added in section 4, and the message of this paper was clarified in section 5 (conclusion).

>The manuscript is globally reasonably well-written yet should be checked for grammatical errors.

This paper got English proofreading again.

>The reference list is curious with the volume number preceding the year of publication, followed by pagination.

We re-checked and followed the MDPI Reference List and Citations Style Guide.